# SLAM on the Hexagonal Grid

**DOI:** 10.3390/s22166221

**Published:** 2022-08-19

**Authors:** Piotr Duszak

**Affiliations:** Institute of Automatic Control and Robotics, Warsaw University of Technology, 02-525 Warsaw, Poland; piotr.duszak@pw.edu.pl

**Keywords:** simultaneous localization and mapping, hexagonal grid, lidar, SLAM

## Abstract

Hexagonal grids have many advantages over square grids and could be successfully used in mobile robotics as a map representation. However, there is a lack of an essential algorithm, namely, SLAM (simultaneous localization and mapping), that would generate a map directly on the hexagonal grid. In this paper, this issue is addressed. The solution is based on scan matching and solving the least-square problem with the Gauss–Newton formula, but it is modified with the Lagrange multiplier theorem. This is necessary to fulfill the constraints given by the manifold. The algorithm was tested in the synthetic environment and on a real robot and is entirely fully suitable for the presented task. It generates a very accurate map and generally has even better precision than the similar approach implemented on the square lattice.

## 1. Introduction

There are three ways to fill a plane with congruent regular polygons: with equilateral triangles, squares, or equilateral hexagons [1]. The second method has many applications, starting with building chain fences through image registration, display, and processing and ending with map representation, path planning, and localization.

However, this way of filling the plane has disadvantages, such as two types of the neighborhood (by vertices or edges; hence, the distance between neighbors is not always equal to one). With the hexagonal lattice, there is no such problem. Furthermore, a regular hexagon has more axes of symmetry than a square. Moreover, hexagons have the highest area to perimeter ratio of all shapes, with which the plane can be filled.

For these reasons, hexagonal grids commonly occur in the natural environment and have many applications. They can be seen in beehives, rock formations, turtle shells, and even on Saturn’s north pole [2] (see Figure 1). This grid has great endurance properties; hence, it is used in materials such as graphene [3]. The human retina (see Figure 1) is also formed in such a way [4]. These inspirations have led to the developments in hexagonal image processing [5]. It is a very promising field of research, but has one big drawback. There are no easily accessible devices to capture or display such images.

Another field in which the hexagonal grid can be useful is mobile robotics. In this case, all advantages can be applied, and there is no impediment (unlike in the previous paragraph) because the map is created based on sensors, such as lidar, which are not based on CCD matrices (unlike most cameras). It was shown that the hexagonal grid could be used as a map representation and has better properties than the square grid [6,7]. However, there is no algorithm to build such a map in real-time, directly on the hexagonal grid.

Currently available solutions e.g., Cartographer ROS (https://google-cartographer-ros.readthedocs.io/, accessed on 16 August 2022) [8] achieve very high accuracy and allow for real-time mapping, but like all popular algorithms to date use a square grid. According to the properties of hexagons (like one type of neighborhood), one may believe that accuracy of the localization and quality of the map built on the hexagonal grid would be even better. However, regardless of accuracy, it is worth investigating the possibility of building such a map to exploit its desirable properties, such as better path-finding. To meet this expectation, an algorithm for simultaneous localization and mapping on a hexagonal grid is presented.

## 2. Related Work

### 2.1. Hexagonal Grid

A large amount of work has been conducted on hexagonal grids. For example, they were tested in terms of geometric properties such as transformations [9,10] or distance between two points [11,12,13]. Furthermore, algorithms for drawing straight lines (analogous to Bresenham’s algorithm) and circles were invented [14,15].

Several articles have focused on hexagonal image processing, as a substantial part of robotics. They describe basic algorithms used in computer vision [5,16], e.g., gradient operators [17], edge detection [18,19,20], morphology operations [20], fast Fourier transform [21], filtering [19,22], and features extractions [23,24,25]. Recently, hexagonal grids have been considered in machine learning frameworks as well [26,27,28]. Although all these papers show the benefits of using hexagonal grids in image processing, there are currently no effortlessly available devices for capturing such images. Therefore their use in this area is limited.

Using hexagonal grids is a relatively new approach in mobile robotics. Most research refers to path planning [29,30,31], which turns out to be shorter and smoother than designated on an ordinary square grid. It was also shown that lines, circles, and splines are better represented on a hexagonal lattice [7]. This is important because obstacles often have such shapes. In [6], algorithms for 2.5D map creation was proposed. This generates a map based on a point cloud of the whole area. Therefore, a critical algorithm is lacking to build a map in real-time, based on subsequent data portions.

### 2.2. SLAM

The problem of simultaneous localization and mapping is essential for mobile robotics. It was first introduced in [32], where they solve the problem using a robot equipped with sonar and so-called beacons to match subsequent sensor readings. They also apply an extended Kalman filter to improve the quality of computed estimations. Since then, scientists and engineers have been intensely interested in this problem, and there are now many solutions worth mentioning.

In [33], the authors presented an algorithm using Rao–Blackwellized particle filters and scan-matching procedures to obtain the objective. In [34], the main idea was to use the Gauss–Newton algorithm to match scans. The approach shown in [8] combines scan-to-submap matching with graph optimization based on the branch and bound algorithm. All solutions mentioned in this paragraph are based on the square grid.

There are also many papers comparing various SLAM algorithms [35,36,37] and their applications, for example, in rescue operations [38].

Recently, there were attempts to use machine learning techniques in simultaneous localization and mapping, e.g., neural networks [39,40,41] and reinforcement learning [42].

Several attempts at SLAM based on hexagonal grids have been described [43,44]. In the former, the authors used a robot equipped with an event-based embedded dynamic vision sensor aimed at the ceiling and particle filter to estimate localization. They used collision sensors located on a bumper ring to notice obstacles. Therefore, it requires some characteristic elements on the ceiling and detects objects only from a very short distance. In the latter, there is a lidar-based algorithm described that relies on matching maps at different moments in time.

The algorithm presented here does not need particular markers. Due to being based on lidar, it can recognize walls and obstacles from a range. Moreover, unlike solutions described in [8,33,34], it uses the hexagonal grid instead of the square one. Similarly, like in [44], it is based on scan matching, but here the entire map is taken into account, not just the edge pixels.

## 3. Localization and Mapping

### 3.1. Problem Formulation

SLAM, as the name suggests, is the problem of creating a map of the environment and estimating the pose of the robot at the same time, based on consecutive sensor readings over time. In this paper, the native map on the hexagonal grid is built, not using the square lattice case. Each cell represents the probability of being occupied. It was assumed that the robot is equipped with lidar because it is one of the most popular sensors used in mobile robotics, and it gives sufficient data for this problem.

Although odometry is also very popular, it was decided not to use such information for the following reasons:Odometry is not very accurate;The system that does not require odometry is more flexible;One of the aims of this article is to compare properties between hexagonal and square lattices.

### 3.2. Solution Overview

The algorithm presented here was inspired by [34], because the solutions described there take into account the fact that a map is built on a grid. It consists of several steps as shown in Figure 2.

First, the robot obtains data from lidar. Readings are in the form $\left(\phi_{i},r_{i}\right)$, where $r_{i}$ is a successive range in the direction of $\phi_{i}$. Therefore, it is necessary to convert them to the proper coordinate system.

The next part is matching the new point cloud to the existing map. It is based on the Gauss–Newton algorithm for solving a non-linear least-square problem. It requires access to every point on the map and calculating derivatives; therefore, a new method must be constructed. This step uses transformation found from the previous reading as the starting point of the Gauss–Newton iteration method.

The last step is the actualization of the map combined with the discretization of the transformed new lidar scan. In the following subsections, all phases are described. However, to make the article more accessible, the unique coordinate system used for the hexagonal grid is first presented.

### 3.3. Coordinate System

In this research, the so-called cube coordinate system is used [9,45]. It was chosen because of the high symmetry level. It has three axes, rotated every 120${}^{\circ }$. It can be interpreted as a coordinate system on a plane with one redundant axis, but also as a Cartesian coordinate system in 3D, but only on a plane, satisfying the equation:(1)x+y+ζ=0

The cube coordinate system is presented in Figure 3. The third axis is denoted by ζ to be consistent with [6] and to distinguish it from the normal third dimension.

### 3.4. Polar to Cube Conversion

Conversion from polar to cube coordinates is analogous to the conversion from the polar to Cartesian system. It can be described by the following formulas:(2)$$\begin{array}{cc} x_{i} & =r_{i}\cdot \cos\left(\phi_{i}\right)\\ y_{i} & =r_{i}\cdot \cos\left(\frac{2}{3}\pi -\phi_{i}\right)\\ \zeta_{i} & =r_{i}\cdot \cos\left(\frac{4}{3}\pi -\phi_{i}\right) \end{array}$$
where:

$r_{i}$, $\phi_{i}$ are the original polar coordinates of the *i*-th point from the lidar scan;

$x_{c}$, $y_{c}$, $\zeta_{c}$ are continuous cube-hexagonal coordinates.

### 3.5. Map Access

The map is stored as a discrete hexagonal grid, but the Gauss–Newton algorithm (which is used in the next step) requires values and gradients everywhere, so it is necessary to interpolate values between cells. It was achieved for both occupancy probability and derivatives by linear interpolation.

For the given point $P_{m}$, with continuous coordinates, the occupancy value $M(P_{m})$ and gradient $\nabla M\left(P_{m}\right)=\left(\frac{\partial M}{\partial x}\left(P_{m}\right),\frac{\partial M}{\partial y}\left(P_{m}\right),\frac{\partial M}{\partial \zeta }\left(P_{m}\right)\right)$ can be computed based on three nearest points with integer coordinates (see Figure 4) as the affine combination according to the formula:(3)$$M\left(P_{m}\right)=\alpha M\left(P_{1}\right)+\beta M\left(P_{2}\right)+\gamma M\left(P_{3}\right)$$
where:

*M* (·) is the occupancy probability;

P1,P2,P3 are the nearest points, shown in Figure 4;

α,β,γ denote the weights of the corresponding points.

In order to calculate weights, it is necessary to solve the following equation:(4)$$\left[\begin{array}{ccc} x_{1} & x_{2} & x_{3}\\ y_{1} & y_{2} & y_{3}\\ \zeta_{1} & \zeta_{2} & \zeta_{3}\\ 1 & 1 & 1 \end{array}\right]\left[\begin{array}{c} \alpha \\ \beta \\ \gamma \end{array}\right]=\left[\begin{array}{c} x\\ y\\ \zeta \\ 1 \end{array}\right]$$

It always has a solution, because points $P_{1}$, $P_{2}$, $P_{3}$ are not collinear and for every point $x_{i}+y_{i}+\zeta_{i}=0$. Taking into account the distribution of a point in the grid, Equation (Equation 4) can be transformed into the following:(5)$$\left[\begin{array}{ccc} x_{1} & x_{1} & x_{1}+1\\ y_{1} & y_{1}+1 & y_{1}\\ \zeta_{1}+1 & \zeta_{1} & \zeta_{1}\\ 1 & 1 & 1 \end{array}\right]\left[\begin{array}{c} \alpha \\ \beta \\ \gamma \end{array}\right]=\left[\begin{array}{c} x\\ y\\ \zeta \\ 1 \end{array}\right]$$
which has a very simple solution:(6)$$\begin{array}{cc} \alpha & =1+\left(x_{1}-x\right)+\left(y_{1}-y\right)\\ \beta & =1+\left(\zeta_{1}-\zeta \right)+\left(x_{1}-x\right)\\ \gamma & =1+\left(y_{1}-y\right)+\left(\zeta_{1}-\zeta \right) \end{array}$$

During calculating derivatives, it is important to remember that variables *x*, *y*, and ζ are dependent on each other. Hence, partial derivatives for α are equal:(7)$$\begin{array}{cc} \frac{\partial \alpha }{\partial x} & =-0.5\\ \frac{\partial \alpha }{\partial y} & =-0.5\\ \frac{\partial \alpha }{\partial \zeta } & =1 \end{array}$$

Similarly, partial derivatives for β and γ can be computed. Hence, derivatives of the occupancy probability are equal:(8)$$\begin{array}{cc} \frac{\partial M}{\partial x}\left(P_{m}\right) & =-0.5\cdot M\left(P_{1}\right)-0.5\cdot M\left(P_{2}\right)+M\left(P_{3}\right)\\ \frac{\partial M}{\partial y}\left(P_{m}\right) & =-0.5\cdot M\left(P_{1}\right)+M\left(P_{2}\right)-0.5\cdot M\left(P_{3}\right)\\ \frac{\partial M}{\partial \zeta }\left(P_{m}\right) & =M\left(P_{1}\right)-0.5\cdot M\left(P_{2}\right)-0.5\cdot M\left(P_{3}\right) \end{array}$$

### 3.6. Scan Matching

In this step of the algorithm, new lidar data are matched to the already existing map. It was inspired by [34], but adapted to the hexagonal grid and cube-coordinate system. This approach is based on fitting beam endpoints to cells on the map, that are already recognized as occupied. This solution was chosen, because it is grid-based, so it can fully exploit the good properties of the hexagonal lattice.

Scan matching process is expected to be more accurate when operating on the hexagonal grid because there is only one type of neighborhood (as mentioned in Section 1) and the average distance from one cell to all neighbors is less for the hexagonal lattice than for the square lattice (if one assumes the surface areas of both polygons is one than the average distance is equal to 1.07 for the former case and it is 1.27 for the latter; see Figure 5).

During this step of algorithm, there is needed the rigid transformation $\xi ={\left(p_{x},p_{y},p_{\zeta },\psi \right)}^{T}$, which minimizes:(9)$$\xi^{*}=\underset{\xi }{\mathrm{argmin}}\sum_{i=1}^{n}{\left[1-M\left(S_{i}\left(\xi \right)\right)\right]}^{2}$$
where $S_{i}$ are the coordinates of the *i*-th scan endpoint after applying transformation ξ. Denote the *i*-th scan endpoint coordinates as $s_{i}={\left(x_{i},y_{i},\zeta_{i}\right)}^{T}$. Then, $S_{i}\left(\xi \right)$ can be computed according to the formula:(10)$$S_{i}\left(\xi \right)=\frac{1}{3}\left[\begin{array}{ccc} 2\cos\psi +1 & 1-\cos\psi -\sqrt{3}\sin\psi & 1-\cos\psi +\sqrt{3}\sin\psi \\ 1-\cos\psi +\sqrt{3}\sin\psi & 2\cos\psi +1 & 1-\cos\psi -\sqrt{3}\sin\psi \\ 1-\cos\psi -\sqrt{3}\sin\psi & 1-\cos\psi +\sqrt{3}\sin\psi & 2\cos\psi +1 \end{array}\right]\left[\begin{array}{c} x_{i}\\ y_{i}\\ \zeta_{i} \end{array}\right]+\left[\begin{array}{c} p_{x}\\ p_{y}\\ p_{\zeta } \end{array}\right]$$

The first part of Formula (Equation 10) is a rotation by angle ψ around the axis, which goes through zero and is perpendicular to the plane, given by the formula x+y+ζ=0; the second part is simply the translation.

Finding ξ is based on the Gauss–Newton method, but it is necessary to find the solution that meets the condition $p_{x}+p_{y}+p_{\zeta }=0$. Therefore, in this paper, the Gauss–Newton method is combined with the method of Lagrange multipliers, which can be used for finding local extremes on manifolds.

Denote $f\left(\xi \right)=\sum_{i=1}^{n}{\left[1-M\left(S_{i}\left(\xi \right)\right)\right]}^{2}$ and $G\left(\xi \right)=p_{x}+p_{y}+p_{\zeta }$. Therefore, according to the Lagrange multiplier theorem finding the minimum of function *f* on manifold G(ξ)=0 requires solving the following equation system:(11)$$\begin{array}{cc} \nabla f(\xi ) & =\lambda \nabla G(\xi )\\ G(\xi ) & =0 \end{array}$$

To calculate partial derivatives of function *f*, the part under the sum is expanded in the Taylor series in the following way:(12)$$f\left(\xi +\Delta \xi \right)≃\sum_{i=1}^{n}{\left[1-M\left(S_{i}\left(\xi \right)\right)-\nabla M\left(S_{i}\left(\xi \right)\right)\frac{\partial S_{i}\left(\xi \right)}{\partial \xi }\Delta \xi \right]}^{2}$$

Now, it is easier to calculate ∇f and substitute it into Equation (Equation 11):(13)$$\begin{array}{cc} 2\sum_{i=1}^{n}{\left[\nabla M\left(S_{i}\left(\xi \right)\right)\frac{\partial S_{i}\left(\xi \right)}{\partial \xi }\right]}^{T}\left[1-M\left(S_{i}\left(\xi \right)\right)-\nabla M\left(S_{i}\left(\xi \right)\right)\frac{\partial S_{i}\left(\xi \right)}{\partial \xi }\Delta \xi \right] & =\left[\begin{array}{c} \lambda \\ \lambda \\ \lambda \\ 0 \end{array}\right]\\ p_{x}+p_{y}+p_{\zeta } & =0 \end{array}$$

Now, this is a system of five equations with five unknowns: $p_{x},p_{y},p_{\zeta },\psi ,\lambda$. Solving it for Δξ yields the Gauss–Newton equation for the minimalization, which takes into account constraints given by manifold:(14)$$\left[\begin{array}{c} \Delta \xi \\ \lambda \end{array}\right]=\left[\begin{array}{c} p_{x}\\ p_{y}\\ p_{\zeta }\\ \psi \\ \lambda \end{array}\right]={\tilde{H}}^{-1}\cdot \left[\begin{array}{c} \sum_{i=1}^{n}{\left[\nabla M\left(S_{i}\left(\xi \right)\right)\frac{\partial S_{i}\left(\xi \right)}{\partial \xi }\right]}^{T}\left[1-M(S_{i}\left(\xi \right))\right]\\ 0 \end{array}\right]$$
where: (15)$$\tilde{H}=\left[\begin{array}{cc} H & \begin{array}{c} 0.5\\ 0.5\\ 0.5\\ 0 \end{array}\\ \begin{array}{cccc} 1 & 1 & 1 & 0 \end{array} & 0 \end{array}\right]$$
with:(16)$$H=\sum_{i=1}^{n}{\left[\nabla M\left(S_{i}\left(\xi \right)\right)\frac{\partial S_{i}\left(\xi \right)}{\partial \xi }\right]}^{T}\left[\nabla M\left(S_{i}\left(\xi \right)\right)\frac{\partial S_{i}\left(\xi \right)}{\partial \xi }\right]$$

In Section 3.5, it was described how to compute *M*(·) and ∇*M*(·). To compute $\frac{\partial S_{i}\left(\xi \right)}{\partial \xi }$, it is necessary to use Equation (Equation 10) to obtain:(17)$$\frac{\partial S_{i}\left(\xi \right)}{\partial \xi }=\left[\begin{array}{cccc} 1 & 0 & 0 & -x_{i}\sin\psi +\frac{\zeta_{i}-y_{i}}{\sqrt{3}}\cos\psi \\ 0 & 1 & 0 & -y_{i}\sin\psi +\frac{x_{i}-\zeta_{i}}{\sqrt{3}}\cos\psi \\ 0 & 0 & 1 & -\zeta_{i}\sin\psi +\frac{y_{i}-x_{i}}{\sqrt{3}}\cos\psi \end{array}\right]$$

When Δξ is calculated, it is now possible to step toward the minimum of function *f*. The starting point of this modified Gauss–Newton iteration formula is transformation calculated based on the previous reading from the lidar.

### 3.7. Map Actualization

After obtaining new data from lidar and determining the new transformation ξ, the map is updated straightforwardly, i.e., beam endpoint coordinates after applying ξ are marked as occupied, and cells between the robot and endpoints are marked as free.

However, due to the discrete character of the hexagonal grid, there is a need to convert continuous cube coordinates to its discrete version. Such an algorithm was described in [9]. For completeness, it is also presented here. The aim is to determine the coordinates of the nearest cell, when continuous coordinates $x_{c},y_{c},\zeta_{c}$ are given. First, the auxiliary variables are calculated:(18)$$\begin{array}{cc} {\hat{x}}_{d} & =\lfloor x_{c}/w\rfloor \\ {\hat{y}}_{d} & =\lfloor y_{c}/w\rfloor \\ {\hat{\zeta }}_{d} & =\lfloor \zeta_{c}/w\rfloor \end{array}$$
where *w* is the width of one cell.

Next, the following Algorithm 1 is applied:
**Algorithm 1** Discretization of Hexagonal Coordinates    **if**
${\hat{x}}_{d}+{\hat{y}}_{d}+{\hat{\zeta }}_{d}=0$
**then**       $x_{d}\leftarrow {\hat{x}}_{d}$       $y_{d}\leftarrow {\hat{y}}_{d}$       $\zeta_{d}\leftarrow {\hat{\zeta }}_{d}$    **else**       **if** $\left\{x_{c}/w\right\}\geq \left\{y_{c}/w\right\}$
**and**
$\left\{x_{c}/w\right\}\geq \left\{\zeta_{c}/w\right\}$ **then**         $x_{d}\leftarrow -{\hat{y}}_{d}-{\hat{z}}_{d}$         $y_{d}\leftarrow {\hat{y}}_{d}$         $\zeta_{d}\leftarrow {\hat{\zeta }}_{d}$    **else if**
$\left\{y_{c}/w\right\}\geq \left\{x_{c}/w\right\}$
**and**
$\left\{y_{c}/w\right\}\geq \left\{\zeta_{c}/w\right\}$
**then**         $x_{d}\leftarrow {\hat{x}}_{d}$         $y_{d}\leftarrow -{\hat{x}}_{d}-{\hat{\zeta }}_{d}$         $\zeta_{d}\leftarrow {\hat{\zeta }}_{d}$       **else**        $x_{d}\leftarrow {\hat{x}}_{d}$         $y_{d}\leftarrow {\hat{y}}_{d}$         $\zeta_{d}\leftarrow -{\hat{x}}_{d}-{\hat{y}}_{d}$       **end if**   **end if**

where {•} denotes the fractional part.

Analogous to [34], to prevent from falling in a local minimum, three maps are kept with different resolutions. Each map has cells with a surface area four times larger than the previous one. During the matching process, first the transformation is found on a coarse map; next, this transformation is used as input for a more accurate map, and so on.

## 4. Results

The algorithm presented here was tested in an artificial environment, simulated by ROS (Robot Operating System) [46] and compared to the Hector SLAM [34], available as a package on ROS, and also on the real mobile robot in the laboratory. Additional tests were conducted on actual data provided by the MIT Stata Center (http://projects.csail.mit.edu/stata/, accessed on 16 August 2022) [47].

### 4.1. Simulation

The simulation was prepared with ROS stage (http://wiki.ros.org/stage, accessed on 16 August 2022). During simulation tests, ground truth of the robot is known, so there is a possibility of comparing the localization precision of the presented algorithm with another one. Several scenarios were prepared, differing in the following properties:Location of obstacles—two types of map (Figure 6);Resolution of built map—two resolutions: area of one cell on the most accurate map equals 0.01 m^2^ or 0.04 m^2^;Angle between initial direction of robot and walls—three angles: 0°, 45°, and 60°;Resolution of lidar—two resolution: 1000 points in one reading or 360 points.

During each test, two parameters were measured: mean square error of position and mean square error of angle. Results for the algorithm presented here and for comparison for hector_mapping (http://wiki.ros.org/hector_mapping, accessed on 16 August 2022) can be seen in Table 1. Test parameters are presented in the following order: map type (maze or rooms), area of cell on the most accurate map (0.01 m^2^ or 0.04 m^2^), angle (0${}^{\circ }$, 45${}^{\circ }$, or 60${}^{\circ }$), and number of points from lidar (1000 or 360).

As can be seen, the precision of the estimated position was always better for the SLAM on the hexagonal grid (usually two times better). Some tests precisions of the estimated angle were slightly better for the hexagonal grid and some for the square grid, but the difference here was minimal. Increasing the cell width twice also doubled the error, which is expected. The angle between the starting direction of the robot and the walls did not affect the results. The algorithm also coped with the reduced resolution of the lidar. Figure 7 shows the generated map for Test 1, and Figure 8 shows the ground truth position of the robot during Test 1 and estimation obtained from hector and hexagonal SLAM.

The algorithm was tested on Ubuntu 20.04.4 LTS, AMD Ryzen 7 4800H CPU 1.4 GHz, RAM 32 GB (manufactured for Lenovo in China). On this hardware, one loop of the algorithm took an average of $1.5\;\times 10^{-2}$ s for a 360-point lidar and $2.9\;\times 10^{-2}$ s for a 1000-point lidar.

### 4.2. Laboratory

The algorithm was also tested on a real mobile robot, i.e., the Husarion ROSbot 2.0 (https://husarion.com/manuals/rosbot/, accessed on 16 August 2022). It is equipped with RpLidar with a 360${}^{\circ }$ field of view and a range of up to 8 m. During experiments in the laboratory, the robot was driven through the maze (see Figure 9) remotely controlled by an operator at an average linear speed of approximately 0.07 m/s and at an average angular speed of approximately 0.02 rad/s. However, this time the accurate position of the robot was not available during the whole run.

To overcome this, the starting and ending positions of the robot were the same (a special marker was attached to the floor for this purpose), and the estimation error was measured after the execution of one full lap. The center of the lidar was established as the reference point of the robot, and the starting and ending position was measured with a caliper relatively to markers attached to the floor. Angle error was not measured due to the inability to determine its ground-truth value, even after one full lap. Two tests were carried out with different arrangements of walls. Final error values were equal to $3.1\;\times 10^{-2}$ m for both tests.

### 4.3. MIT Stata Center Data Set

MIT Stata Center Data Set was collected in buildings belonging to MIT in 2011 and 2012. It contains information about odometry, cameras recordings, and—most importantly for this research data—from lidar. The authors used Willow Garage PR2 equipped with Hokuyo UTM-30LX Laser Scanner. Moreover, they share ground truth positions for some recordings with declared accuracy of 2–3 cm.

Two tests were provided, differing in their starting position and a robot’s motion trajectory. During the first one (from 27 January 2012) the robot was driving at an average linear speed of 0.57 m/s (a maximum linear speed was 1.28 m/s) and at an average angular speed of 0.11 rad/s (a maximum angular speed was 0.70 rad/s). Speeds during the second one (from 3 April 2012) were similar, namely: an average linear speed of 0.54 m/s, a maximum linear speed of 1.10 m/s, an average angular speed of 0.15 rad/s, and a maximum angular speed of 0.78 rad/s.

Figure 10 shows the ground truth position of the robot during the first test, using MIT Data and estimations obtained from hector and hexagonal SLAM. Analogous results for the second one are shown in Figure 11. Those experiments confirm the results obtained from the simulations, i.e., the trajectory determined by the hexagonal SLAM runs closer to the real one than the trajectory determined by hector SLAM. Furthermore, during both tests, hector SLAM got lost at some point while the hexagonal SLAM did not.

## 5. Conclusions and Future Works

In this paper, the algorithm for simultaneous localization and mapping directly on a hexagonal grid was presented. The algorithm generates a very accurate map, which can be used, for example, for path planning. Therefore, all advantages of the hexagonal grid can now be used in mobile robotics, which is the main achievement of this article. Additionally, it was shown that localization on the hexagonal grid is more accurate than on the square grid.

In the future, this algorithm will be improved by adding probabilistic aggregation of the map and information given by the odometry, and by applying Kalman or particle filter to estimate movement parameters. Moreover, loop closure will be added, and other methods of map access will be tested, including other types of interpolation or using more than three nearest points.

## Figures and Tables

**Figure 1 sensors-22-06221-f001:**
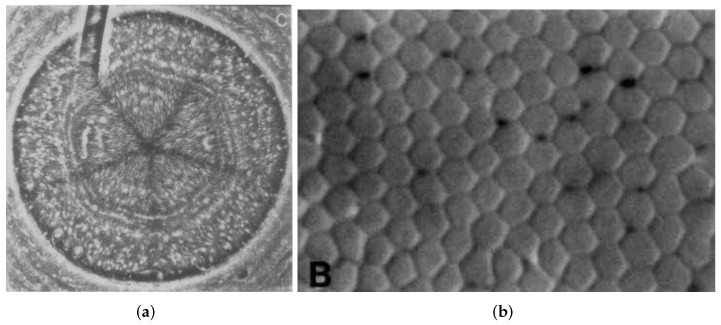
Hexagonal structures occurring in nature. (**a**) Saturn’s north pole [4]. (**b**) Cross-section of human retina [2].

**Figure 2 sensors-22-06221-f002:**
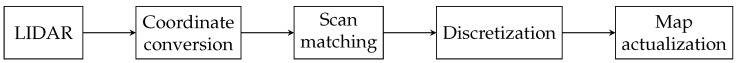
Overview of the algorithm.

**Figure 3 sensors-22-06221-f003:**
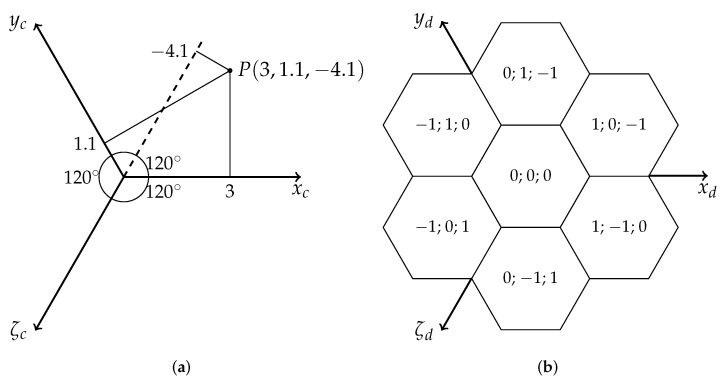
Cube coordinate system. (**a**) Continuous space with example point. (**b**) Discrete grid.

**Figure 4 sensors-22-06221-f004:**
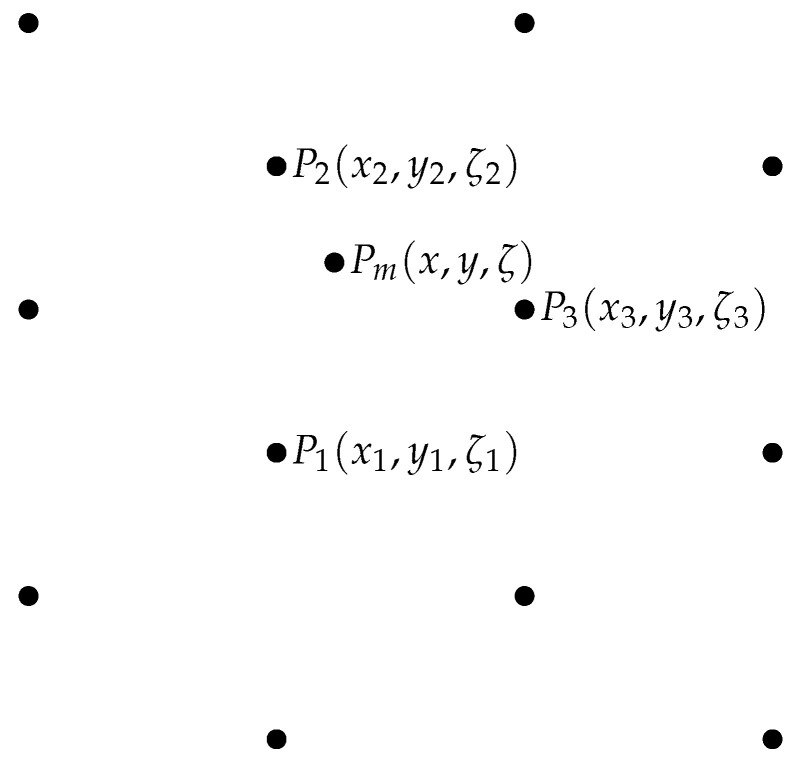
Point $P_{m}$ with continuous coordinates and its nearest integer neighbors $P_{1}$, $P_{2}$, $P_{3}$.

**Figure 5 sensors-22-06221-f005:**
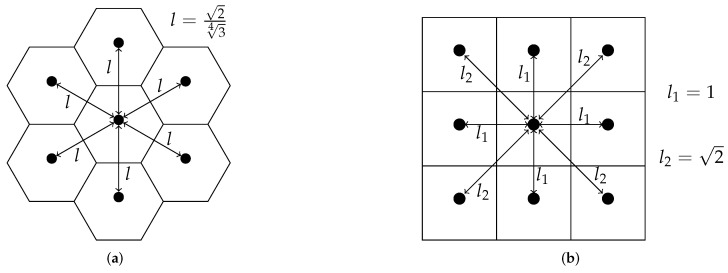
Comparison of the neighborhood for square and hexagonal lattice. Distances given when area of polygons is one. (**a**) The distance to all neighbors is the same at approximately 1.07. (**b**) There are two types of neighborhood. Avarage distance to the neighbor is approximately 1.21.

**Figure 6 sensors-22-06221-f006:**
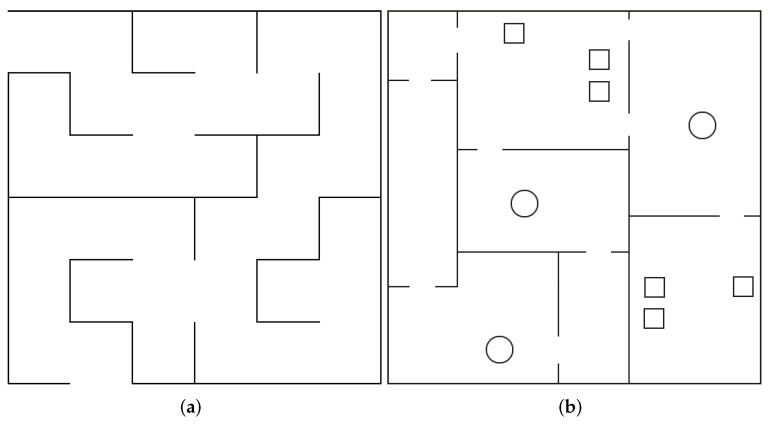
Two types of synthetic maps. (**a**) Maze from robotcraft competition (https://robotcraft.ingeniarius.pt/, https://github.com/ingeniarius-ltd/robotcraft_maze, accessed on 16 August 2022). (**b**) Synthetically generetaed rooms.

**Figure 7 sensors-22-06221-f007:**
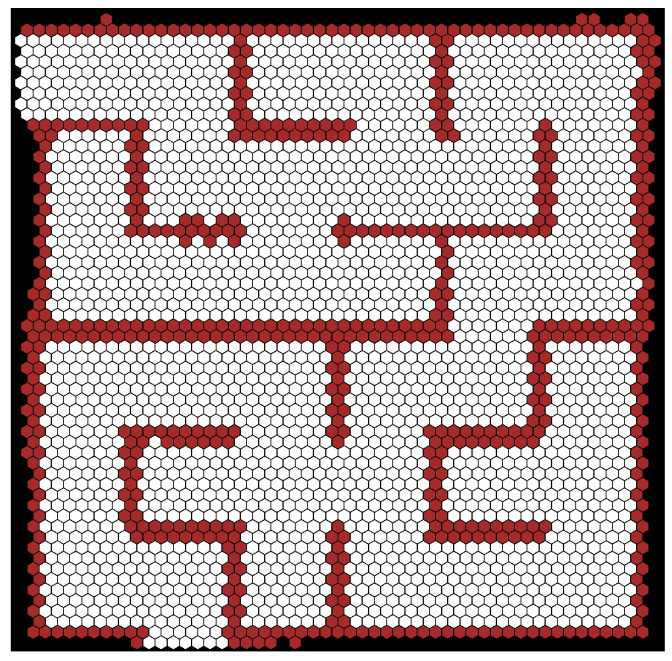
Map of the maze generated by the algorithm.

**Figure 8 sensors-22-06221-f008:**
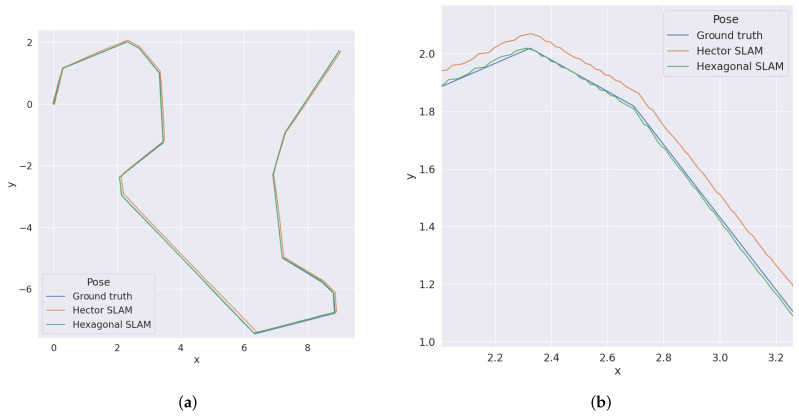
Ground truth position and position estimations for Test 1. (**a**) The whole test run. (**b**) Close up.

**Figure 9 sensors-22-06221-f009:**
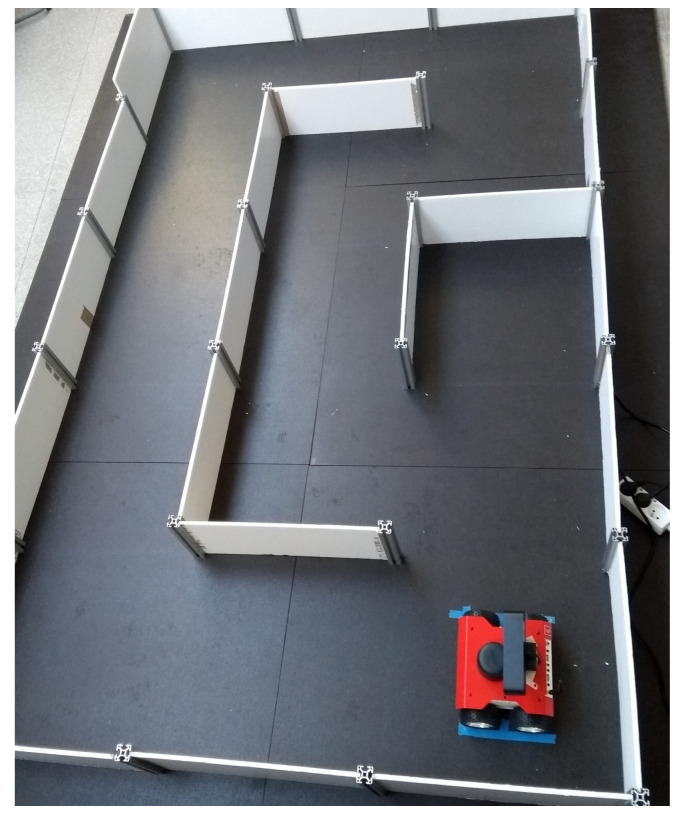
Husarion ROSbot in the maze.

**Figure 10 sensors-22-06221-f010:**
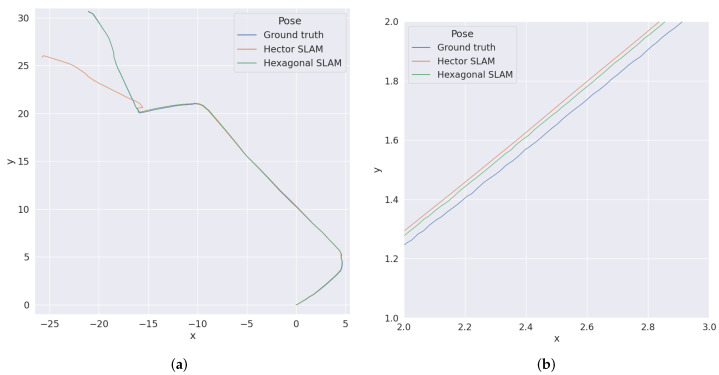
Ground truth position and position estimations for MIT Data Set—recording from 27 January 2012. (**a**) The whole test run. (**b**) Close up.

**Figure 11 sensors-22-06221-f011:**
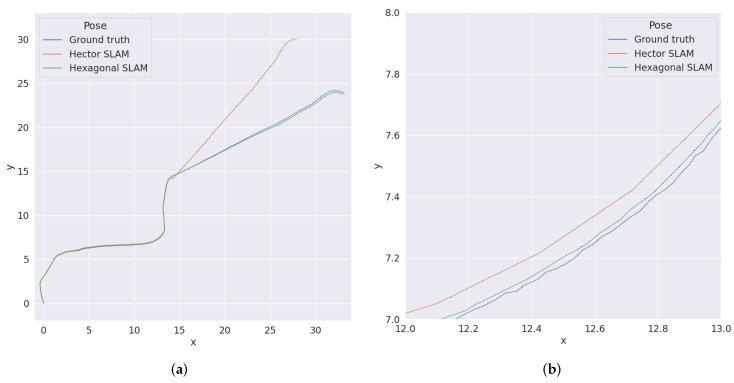
Ground truth position and position estimations for MIT Data Set—recording from 3 April 2012. (**a**) The whole test run. (**b**) Close up.

**Table 1 sensors-22-06221-t001:** Localization error for Hexagonal SLAM and hector_slam for various situations.

		MSE of	MSE of	MSE of Angle	MSE of Angle
Test	Test	Position	Position for	for	for
Number	Parameters	for Hexagonal	Hector_slam	Hexagonal	Hector_slam
		SLAM	in Meters	SLAM	in Radians
		in Meters		in Radians	
	maze				
1	0.01 m^2^	**2.4 × 10^−2^**	6.1 × 10^−2^	1.8 × 10^−4^	**1.3 × 10^−4^**
	0°				
	1000				
	rooms				
2	0.01 m^2^	**5.8 × 10^−2^**	6.3 × 10^−2^	8.0 × 10^−5^	**6.9 × 10^−5^**
	0°				
	1000				
	maze				
3	0.04 m^2^	**5.01 × 10^−2^**	1.4 × 10^−1^	**1.7 × 10^−4^**	3.4 × 10^−4^
	0°				
	1000				
	maze				
	maze				
4	0.01 m^2^	**2.3 × 10^−2^**	6.0 × 10^−2^	**2.4 × 10^−4^**	2.6 × 10^−4^
	45				
	1000				
	maze				
5	0.01 m^2^	**1.5 × 10^−2^**	4.1 × 10^−2^	**1.7 × 10^−4^**	4.2 × 10^−4^
	60				
	1000				
	maze				
6	0.01 m^2^	**2.4 × 10^−2^**	5.3 × 10^−2^	2.0 × 10^−4^	**1.2 × 10^−4^**
	0				
	360				

## Data Availability

The source code and data used to support the findings of this study are available from the corresponding author upon request.

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
