# Peer review of "SLAM on the Hexagonal Grid"

_sensors, 2022, doi:10.3390/s22166221_

Round 1

Reviewer 1 Report

The authors proposed hexagonal grid for SLAM instead of using the traditional square grid. The core component of the algorithm is scan matching with hexagonal grid. The algorithm was tested in a simulation and a real indoor environment. My main concern is the operation in hexagonal grid is computational cost, while the improvement of localization is insignificant when compared to the square grid if a small grid size is used. Some of my detailed comments can be found below:
1. In the Introduction section, it is important to show what is the state-of-the-art. For the reviewer’s point of view, it is not clear if the investigation of hexagonal is necessary.
2. The paper is talking about a SLAM solution based on hexagonal grid. But Fig. 2 does not include any loop closure detection and optimization, which are two important modules in most popular SLAM systems.
3. In Fig. 4, only three nearest points are used for map access? Will you get a get a better accuracy with more nearest points?
4. Scan matching is a key module in the paper. The authors should explain why this hexagonal grid can improve the accuracy when compared to square grid-based scan matching.
5. How is the computation involved in the proposed approach. The authors should add analysis and discussion on this point.
6. Maybe you can show or compare the trajectory based on Hector SLAM and the proposed hexagonal SLAM.
7. In the conclusion, it is not scientifically correct to say “it meets all expectations”, as a thorough comparison in the paper is missing. It is also important to point out the weakness of the proposed scheme.

Reviewer 2 Report

The paper introduces a novel algorithm to deal with SLAM within an hexagonal grid.

In general, I found that writing needs a significant improvement as many parts are not detailed enough. To provide some examples:

- The introduction is not providing the context for this work. Despite why the hexagonal grid is somehow justified, I miss more details in the objectives, motivation and main contributions of this work. Is it expected a huge gain with the hexagonal grid to justify its election?

- The related work cites many works, but I miss a good review (not just one paragraph) on, for instance, how hexagonal grids were applied to robotics. The same applied to the related work on SLAM, I miss a focused review on SLAM and grids (including hexagonal). In general, author is not providing any summary of the related work with the identified gaps in research and the address he is addressing them on this paper. Bold sentences (such as "So there is a great difference between their algorithm and the one described here.") must be avoided if author does not provide evidence or justification. 

- Could author please highlight the novelty of the proposed method with respect to the literature on SLAM?

- In general, the experimental setup and results are not properly described. The results are provided without any further discussion, real test are limited to a very small area. The way to obtain the ground truth is not provided and angles could not be measured. 

- As a major point, I consider that the quality of presentation (including writing style) is not mature enough for publication in MDPI Sensors. 

Minor:

- Check grammar. e.g. "machine learn" is a concept that does not exist. 

In the simulated results, please highlight the best result for each test number.  

Round 2

Reviewer 1 Report

All my comments have been addressed by the authors.

Author Response

Thank you very much.

Reviewer 2 Report

I still consider some core points are not addressed.

In general, the experimental setup and results are not properly described. For instance, the speed of the testing vehicle in the real-world experiments is not provided. How was the vehicle driven? Was speed constant? Will the system work similarly at different vehicle velocities?

The results are provided without any further discussion, real test are limited to a very small area. As mentioned by the authors, only one full lap is considered. Despite two configurations are performed, the quantity of real-world experiments is limited. 

The way to obtain the ground truth is not provided and angles could not be measured. Still, authors do not describe how the positioning error is calculated. How are those 3.1 cm measured? Where is the reference point for the mobile robot? How this position is calculated with respect to the starting point? This is of utmost relevance as the maximum error in measuring the ground truth should be 3 mm according to the ISO18305. 

Figures still have typos: e.g. Grand truth instead of Ground truth

Author Response

This manuscript is a resubmission of an earlier submission. The following is a list of the peer review reports and author responses from that submission.